# Non-Autoregressive Sentence Ordering

**Yi Bin[2], Wenhao Shi[1], Bin Ji[2], Jipeng Zhang[3], Yujuan Ding[4], Yang Yang[1]**

[1]University of Electronic Science and Technology of China, [2]National University of Singapore
[3]The Hong Kong university of Science and Technology, [4]The Hong Kong Polytechnic University
yi.bin@hotmail.com, jibin@nus.edu.sg
{shiwenhao16, zjp1191299782, dingyujuan385, dlyyang}@gmail.com

## Abstract

Existing sentence ordering approaches generally employ encoder-decoder frameworks with the pointer net to recover the coherence by recurrently predicting each sentence step-by-step. Such an autoregressive manner only leverages unilateral dependencies during decoding and cannot fully explore the semantic dependency between sentences for ordering. To overcome these limitations, in this paper, we propose a novel Non-Autoregressive Ordering Network, dubbed *NAON*, which explores bilateral dependencies between sentences and predicts the sentence for each position in parallel. We claim that the non-autoregressive manner is not just applicable but also particularly suitable to the sentence ordering task because of two peculiar characteristics of the task: 1) each generation target is in deterministic length, and 2) the sentences and positions should match exclusively. Furthermore, to address the repetition issue of the naive non-autoregressive Transformer, we introduce an exclusive loss to constrain the exclusiveness between positions and sentences. To verify the effectiveness of the proposed model, we conduct extensive experiments on several common-used datasets and the experimental results show that our method outperforms all the autoregressive approaches and yields competitive performance compared with the state-of-the-arts. The codes are available at: https://github.com/steven640pixel/nonautoregressive-sentence-ordering.

## 1 Introduction

Sentence ordering is one of the fundamental and common tasks to model document coherence, which targets at re-organizing a set of sentences into a coherent paragraph (as shown in Figure 1). Most early works of sentence ordering (Lapata, 2003; Barzilay and Lee, 2004; Barzilay and Lapata, 2008) apply probabilistic transition model and rule-based model, *e.g.*, HMM, entity, and content model, based on hand-crafted features. While such

Figure 1: An example of sentence ordering sourced from SIND dataset. The goal of sentence ordering is to understand the semantics and logic of a set of unordered sentences (the left box), and reorganize them to a coherent paragraph (the right box).

sophisticated designs are costly in labor and time, expertise-required, and even cannot be well generalized to other scenarios. In the past couple of years, inspired by the great success of deep learning, dozens of deep neural methods for sentence ordering have been proposed and achieved great success (Cui et al., 2018; Kumar et al., 2020; Wang and Wan, 2019; Prabhumoye et al., 2020; Basu Roy Chowdhury et al., 2021).

There exist two paradigms of neural sentence ordering: 1) ranking model via the scores of relative positions between paired sentences, and 2) generation model based on an encoder-decoder framework to predict the sentence order with sequence generation. The former one, ranking model (Prabhumoye et al., 2020; Chen et al., 2016), starts from calculating a score for each sentence pair to indicate their relative position in a coherent paragraph, then employs ranking or searching strategies to derive out the gold order. Since the score between paired sentences only measures the one-to-one relative position, which may fail to capture the one-to-many interactions on the global context level. Besides, the naive ranking process is somehow brute-force and time-consuming, and highly depends on the quality of sentence representation.

The latter one, generation model regards sentence ordering as a sequence prediction problem to explore the *relations between sentences and positions*, which takes a set of sentences and generates

the coherent sentence sequence. At the early stage, several encoder-decoder based approaches (Gong et al., 2016; Logeswaran et al., 2018) treat the unordered input sentences as a permutation sequence and encode them sequentially. Such sequential modeling encodes incorrect sentence order and semantic logic between sentences, which may mislead the decoder to predict an incoherent paragraph. Specifically, with sequential modeling, different permutations of the same paragraph may derive different paragraph representations and result in different output sentence order, which is not reasonable and inconsistent with the intuition of humans. To address this issue, Cui *et al.* (2018) first propose a deep attentive sentence ordering network equipped with self-attention to learn a reliable and consistent paragraph representation. Their model implements order-independent encoding with a non-positional variant of Transformer (Vaswani et al., 2017), but still predicts sentences via a pointer network as most previous works do. Nevertheless, the RNN-based pointer networks predict the sentence order step-by-step, which only explores the unilateral dependencies with past predictions and fails to leverage the comprehensive bilateral interactions between sentences for ordering.

To overcome the above limitations, in this paper, we propose a novel **N**on-**A**utoregressive **O**rdering **N**etwork, dubbed *NAON*, which explores the bilateral dependencies between sentences and predicts the sentence for each position in parallel. Specifically, our NAON consists of a basic sentence encoder, an order-independent contextual sentence encoder and a non-autoregressive Transformer (NAT) (Gu et al., 2018) decoder. We first employ a sequence encoder, *e.g.*, BERT in this work, to map discrete words in each sentence to a compact representation. The obtained sentence representations are injected to a Transformer (Vaswani et al., 2017) encoder to exploit the interaction and relation between sentences, and export contextual sentence representations. Note that we remove the positional embedding in the Transformer to achieve the order-invariant encoding for the sentences. In contrast to obtaining the paragraph representation via pooling to initialize the memory of auto-regressive decoder (Cui et al., 2018), we make full use of contextual sentence representations through multi-head attention in non-autoregressive decoder. We design a non-autoregressive decoder (NAD) that takes positions as input and predicts

the sentence for each position utilizing a pointer network (Vinyals et al., 2015). In particular, we first employ multi-head attention to explore the relations between positions. Then the output distribution over the original sentence representation set for each position will be modelled by a pointer network. As mentioned in (Gu et al., 2018), naive non-autoregressive decoding suffers from severe problem of output repetition due to the complete *conditional independence*. To address this issue, we elaborate an exclusive loss constraining the exclusiveness between each sentence and position pair during training, and a greedy selective and removing strategy for inference. In summary, our main contributions are as follows:

- We propose a novel approach, non-autoregressive ordering network (NAON), for sentence ordering, which implements self-attention to explore the bilateral interactions between sentences, going beyond the unilateral dependencies of RNN-based step-by-step decoding. To the best of our knowledge, our approach is one of the first attempts using non-autoregressive decoders for the sentence ordering problem.

- To alleviate the repetition problem of vanilla non-autoregressive models, we design an exclusive loss under the implication of exclusive constraint between positions and sentences during training. For inference, we simply design a *greedy selective and removing strategy* to match each sentence-position pair.

- Extensive experiments are conducted on several common-used datasets, and the results demonstrate the effectiveness of the proposed approach.

## 2 The Proposed Approach

### 2.1 Preliminary

Sentence ordering aims to capture the coherence and recover the gold order of a set of unordered sentences. Specifically, given a sentence set $S = \{s_1, s_2, ..., s_N\}$ with $N$ sentences, and each sentence $s_i = [w_{i1}, w_{i2}, ..., w_{iL_i}]$ contains a sequence of $L_i$ words. The goal of sentence ordering is to find an order $O^* = [o_1^*, o_2^*, ..., o_N^*]$ conveying the coherent semantics, which can be modelled by:

$$P(O^*|S) \geq P(O|S), \forall O \in \Omega, \quad (1)$$

where $P(O|S)$ is the probability of order $O$ given $S$, and $\Omega$ denotes the exhaustive set of all potential

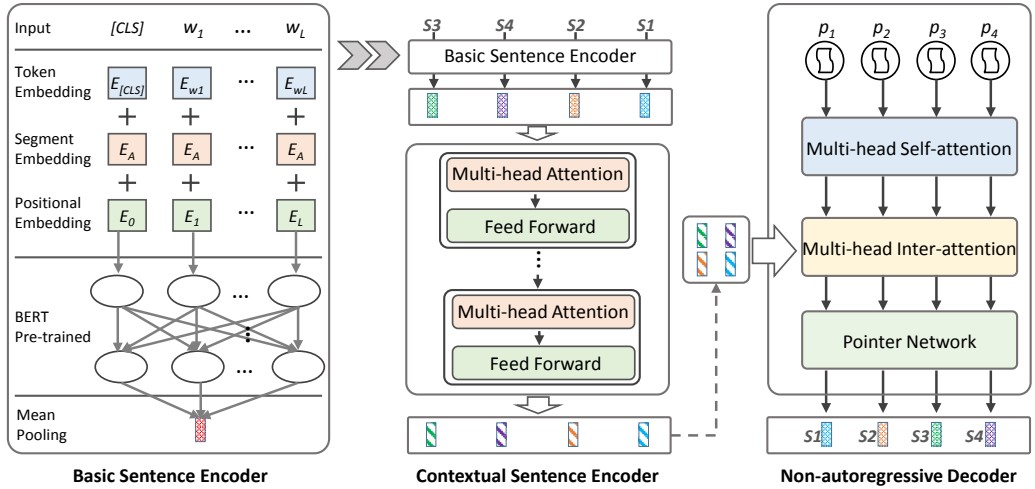

Figure 2: The overall flowchart of the proposed **N**on-**A**utoregressive **O**rdering **N**etwork (NAON), which consists of a basic sentence encoder, a contextual sentence encoder and a non-autoregressive decoder.

order with the size of $A_N^N$. Existing autoregressive approaches (Cui et al., 2018; Wang and Wan, 2019) generate the coherent paragraph sentence-by-sentence to recover the gold order, and optimize the model by maximizing:

$$\sum_{j=1}^{N} \log P(s_j|s_1, s_2, ..., s_{j-1}, S). \qquad (2)$$

While our non-autoregressive ordering network implements the optimization in parallel by maximizing the logarithmic probabilities for all sentences:

$$\sum_{j=1}^{N} \log P(s_j|S). \qquad (3)$$

## 2.2 Basic Sentence Representation

To map discrete words of a sentence into compact representations, as shown on the left of Figure 2, a sequential encoder is first employed, *e.g.*, an LSTM or BERT. Inspired by the success in language modeling of BERT (Devlin et al., 2019), we adopt a pre-trained BERT-base model for basic sentence representation. In particular, following previous work (Kumar et al., 2020), we concatenate the [CLS] token and word sequence as BERT encoder input and average the outputs of all the tokens to denote the entire sentence representation. To make the representation more compatible with sentence ordering task, we also fine-tune it in an end-to-end fashion on each dataset.

## 2.3 Contextual Sentence Representation

Most existing autoregressive approaches (Cui et al., 2018; Wang and Wan, 2019; Logeswaran et al.,

2018) provide a "paragraph representation" to initialize the hidden state and memory cell of the decoder. A straightforward way is assuming that the given order is a "pseudo order" of the unordered sentences, and implementing an RNN-based encoder (Wang and Wan, 2019), *e.g.*, LSTM or GRU, to map sentence representations to a dense feature vector for the paragraph. While such pseudo order assumption introduces incorrect sentence order, which may induce semantic incoherence and mislead the decoder in recovering the ground-truth order. To address this issue, Cui et al. (2018) proposed a deep attentive ordering network integrating a self-attention mechanism to derive the order invariant sentence representations, and adopted mean pooling across sentences to obtain the paragraph representation. Different from existing autoregressive approaches which need to initialize the decoder memory, our non-autoregressive decoder does not require such "paragraph representation". We design a contextual sentence encoder to interact each sentence with all the sentences in the set and injects all the contextual sentence representations into the decoder to find the correct order.

Obviously, it is important to exploit semantic relations between sentences for recovering coherence. Towards this end, we design a contextual sentence encoder, shown in the middle of Figure 2, to interact each sentence with other sentences. Specifically, after obtaining the basic sentence representations $E^b = \{e_1^b, e_2^b, ..., e_N^b\}$, we employ a Transformer-based architecture similar to (Cui et al., 2018), equipped with multi-head self-attention mechanism without the positional embedding, to exploit the

contextual information as follows:

$$\text{Attention}(Q, K, V) = softmax(\frac{QK^T}{\sqrt{d_k}})V, \quad (4)$$

$$\text{MH}(Q, K, V) = [H_1, H_2, ..., H_h]W, \quad (5)$$

$$H_i = \text{Attention}(QW_i^Q, KW_i^K, VW_i^V). \quad (6)$$

To explore deep contextual interaction, we duplicate the multi-head self-attention block multiple times, *e.g.*, 4 for NSF and arXiv datasets in this work. Finally, the contextual sentence representations $E^c = \{e_1^c, e_2^c, ..., e_N^c\}$ can be derived from the output of the last layer.

### 2.4 Non-Autoregressive Prediction

Almost all the previous generation approaches (Logeswaran et al., 2018; Cui et al., 2018; Basu Roy Chowdhury et al., 2021) adopt a recurrent-based architecture composing with a pointer network (Vinyals et al., 2015) to predict the ordered sentence step-by-step, which only leverages the unilateral dependencies with the past prediction, and cannot capture the coherent dependencies between all the sentences. Besides, as argued in (Bengio et al., 2015), the discrepancy between training and inference of RNN-based generation strategy may decrease the inference performance. Inspired by the potentials of the non-autoregressive transformer in neural machine translation, we devise a non-autoregressive decoder (NAD) for recovering the correct order of sentences, with the exploration of bilateral dependencies. Our NAD is designed based on a Non-Autoregressive Transformer (NAT) decoder in machine translation, which removes the autoregressive connections between steps and generates all the target words in parallel, rather than step-by-step (*i.e.*, word-by-word). Beyond the aforementioned superiority of NAT in translation, NAD is especially suitable for sentence ordering from two aspects: (1) the length of the decoder is determined by the input set of sentences, and (2) the sentences and positions match exclusively.

As suggested in (Gu et al., 2018), different inputs of non-autoregressive decoder may lead to quite different outputs resulting in a remarkable performance gap. Thus it is crucial to choose a proper information source for the input of NAD. Towards this end, we meticulously adopt positional information as the input of our non-autoregressive decoder. Because the sentence ordering problem could be interpreted as matching the unordered sentences to the right positions, and it is straightforward to

take the position information as input and predict the correct sentence for each position in parallel. Specifically, we follow previous work (Vaswani et al., 2017) to project each discrete position into a compact representation $P$ via:

$$p_{i,2j} = sin(i/10000^{2j/d_k}), \quad (7)$$

$$p_{i,2j+1} = cos(i/10000^{2j/d_k}), \quad (8)$$

where $i$ denotes the position and $j$ is the $j$-th dimension in $p_i$. We then exploit the interaction and relative information between positions implementing a multi-head self-attention as:

$$\overline{P} = \text{MHAtt}(PW^Q, PW^K, PW^V), \quad (9)$$

where MHAtt denotes multi-head self-attention block comprised of Equation 4-6. To connect the positions and the elements in the unordered context set, we further implement multi-head inter-attention between positions and sentences to exploit the correlations between them. Specifically, we query the contextual sentence representation set using $\overline{P}$, to obtain the attentive representations of sentences as:

$$E^p = softmax(\frac{\overline{P}(E^c)^T}{\sqrt{d_k}})E^c, \quad (10)$$

where $E^c$ is the contextual sentence representation derived from the contextual encoder. With such an operation, the obtained $E^p$ now is position sensitive representation, and could be used to select the most relevant sentence for each position. Specifically, the probability of the sentence $j$ to be in the position $i$ can be obtained by:

$$\omega_{ij} = u^T \tanh(W_p e_i^p + W_b e_j^b), \quad (11)$$

$$Ptr_i = softmax(\omega_i), \quad (12)$$

where $W_*$ are learned parameters, and $u$ is a column weight vector. $e_j^b$ and $e_j^p$ are the basic and positional sensitive representations of sentence $j$. $Ptr_i$ is the probabilistic distribution across all the sentences for $i$-th position.

### 2.5 Training and Inference

**Training:** Following previous ordering works (Cui et al., 2018; Chen et al., 2016), the entire model could be trained to maximize the coherence probability by minimizing the cross entropy loss as:

$$L_c = -\frac{1}{N}\sum_{i=1}^N \log P(o_i|p_i; \Theta), \quad (13)$$

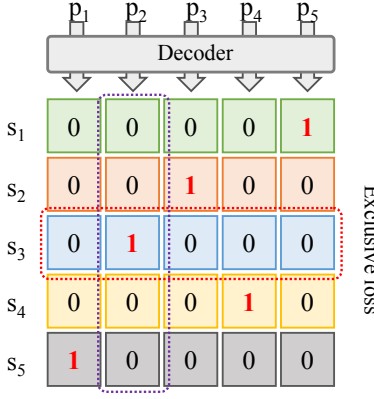

Figure 3: Illustration of our proposed exclusive loss, which simultaneously constrain the sentence selection and position matching.

where $o_i$ is the ground-truth sentence for given $i$-th position $p_i$, and $\Theta$ denotes all the learnable parameters in the model. As each sentence in the ordering problem is allowed to be selected only once, to avoid repeat selection, the RNN-based autoregressive model adopts a mask to remove selected sentences (Yin et al., 2020). However, the non-autoregressive model suffers from the repetition problem for its parallel decoding fashion (Gu et al., 2018). We have observed that the traditional pointer loss only penalizes the repetition for sentence choosing. Considering the characteristic of the ordering task, as depicted in Figure 3, every position should also choose one and only one sentence. Towards this end, we introduce an elegant **exclusive loss** to constrain the mutual exclusiveness of optimal matching between positions and elements. Specifically, we simultaneously calculate two pointers for sentence choosing and position choosing when we obtain the pointer map by Equation 11. Given the pointer map $\omega$, we implement softmax across column and row similar to Equation 12, where the column one interprets choosing sentence for each position and the row one assigns a position for each sentence. Then we optimize the entire model by minimizing the combined exclusive loss as follows:

$$L_{ex} = -\frac{1}{N}\sum_{i=1}^{N}(\log P(o_i|p_i) + \log P(p_i|o_i)). \tag{14}$$

Through such an exclusive constraint, the repetition problem could be distinctly alleviated.

**Inference**: Obviously, the proposed exclusive loss cannot completely avoid repetition problem during decoding. Similar with the masking strategy used in recurrent pointer networks, we introduce a simple yet effective **greedy selective and removing strategy** to ensure the exclusive matching for inference. Given the probability distribution array, as described in Algorithm 1, the proposed strategy greedily selects the largest probability and returns its index indicating the matching position and sentence. It then removes all the probability values of the corresponding column and row by setting them to zero. The process is repeated until all the values of the array are set to zero, which means that the matching process between positions and sentences is completely finished.

---

**Algorithm 1** Pseudo-code of greedy selective and removing in a PyTorch-like style.

---
```
# Ptr: N × N output probability distribution array of NAD
INPUT Ptr
output = []
while sum(Ptr)>0:
    row_idx, col_idx = max_index(Ptr)
    # max_index( ): get the index of the maximum value
    output.append((row_idx, col_idx))
    Ptr[row_idx, :] = 0
    Ptr[:, col_idx] = 0
RETURN output
```
---

## 3 Experiments

To verify the effectiveness of our NAON, we conduct extensive experiments on six datasets, and evaluate the performance with **Acc**, **PMR**, and $\tau$. More detailed experimental settings are described in the Appendix B.

### 3.1 Comparing with Baselines

As aforementioned, existing works lie in two categories: ranking-based and generation-based methods. To assess the proposed NAON, we compare it with the recent state-of-the-art methods, including **Pairwise** model via pairwise ranking (Chen et al., 2016), **RankTxNet** via deep attentive ranking (Kumar et al., 2020), and **B-TSort** searching correct order with topological sort (Prabhumoye et al., 2020) of ranking model, and the first architecture using pointer network **LSTM-Ptr** (Gong et al., 2016), **SetLSTM** designing an LSTM based attention to process the set encoding (Logeswaran et al., 2018), **AON** with a deep attentive order invariant paragraph encoder (Cui et al., 2018), **HAN** equipped with hierarchical attention (Wang and Wan, 2019), **TGCM** modeling topic-guide coherence (Oh et al.,

Table 1: Comparison with baselines. The best and 2nd-best results are in bold and underlined, respectively. NAON-NE indicates no exclusive loss. Besides, we also take an attempt at relative order exploration and BART enhancement, namely NAON-RO and BART-NAON respectively, and compare them with BERSON and RE-BART, listed in the last two blocks in **blue**.

| Model | NIPS | | | AAN | | | NSF | | | arXiv | | | SIND | | | ROCStory | | |
|---|---|---|---|---|---|---|---|---|---|---|---|---|---|---|---|---|---|---|
| | Acc | PMR | $\tau$ | Acc | PMR | $\tau$ | Acc | PMR | $\tau$ | Acc | PMR | $\tau$ | Acc | PMR | $\tau$ | Acc | PMR | $\tau$ |
| Pairwise | - | - | - | - | - | - | - | - | - | - | 33.43 | 0.66 | - | - | - | - | - | - |
| LSTM-Ptr | 50.87 | - | 0.67 | 58.20 | - | 0.69 | 32.45 | - | 0.52 | - | 40.44 | 0.72 | - | 12.34 | 0.48 | - | - | - |
| SetLSTM | 51.55 | - | 0.72 | 58.06 | - | 0.73 | 28.33 | - | 0.51 | - | - | - | - | - | - | - | - | - |
| AON | 56.09 | - | 0.72 | 63.24 | - | 0.73 | 37.72 | - | 0.55 | - | 42.19 | 0.73 | - | 14.01 | 0.49 | - | - | - |
| HAN | - | - | - | - | - | - | - | - | - | - | 44.55 | 0.75 | - | 15.01 | 0.50 | - | 39.62 | 0.73 |
| SE-Graph | 57.27 | - | 0.75 | 64.64 | - | 0.78 | - | - | - | - | 44.33 | 0.75 | - | 16.22 | 0.52 | - | - | - |
| En-Ptr | - | - | - | - | - | - | - | - | - | - | 46.58 | 0.77 | - | 17.37 | 0.53 | - | 46.00 | 0.77 |
| TGCM | 59.43 | 31.44 | 0.75 | 65.16 | 36.69 | 0.75 | **42.67** | **22.35** | 0.55 | 58.31 | 44.28 | 0.75 | 38.71 | 15.18 | 0.53 | - | - | - |
| RankTxNet | - | 24.13 | 0.75 | - | 39.18 | 0.77 | - | 9.78 | 0.58 | - | 43.44 | 0.77 | - | 15.48 | 0.57 | - | 38.02 | 0.76 |
| B-TSort | 61.48 | 32.59 | **0.81** | 69.22 | **50.76** | **0.83** | 35.21 | 10.44 | **0.66** | - | - | - | 52.23 | 20.32 | **0.60** | - | - | - |
| NAON-NE | 63.37 | 32.13 | 0.79 | 68.33 | 45.87 | 0.80 | 40.88 | 12.86 | 0.63 | 63.12 | 45.71 | 0.77 | 51.40 | 18.45 | 0.58 | 74.45 | 52.68 | 0.81 |
| NAON | **64.43** | **33.02** | 0.80 | **69.28** | 46.71 | 0.82 | 41.82 | 13.68 | 0.64 | **64.50** | **46.82** | 0.79 | **52.38** | 19.13 | **0.60** | **75.89** | **53.36** | **0.82** |
| BERSON | 73.87 | 48.01 | 0.85 | 78.03 | 59.79 | 0.85 | 50.02 | 23.07 | 0.67 | 75.08 | 56.06 | 0.83 | 58.91 | 31.69 | 0.65 | 82.86 | 68.23 | 0.88 |
| NAON-RO | 74.15 | 48.03 | 0.85 | 78.22 | 59.81 | 0.86 | 50.39 | 23.11 | 0.67 | 75.16 | 56.12 | 0.83 | 59.04 | 31.71 | 0.66 | 83.07 | 68.31 | 0.88 |
| RE-BART | 77.41 | 57.03 | 0.89 | 84.28 | 73.50 | 0.91 | 50.23 | 29.74 | 0.76 | 74.28 | 62.40 | 0.86 | 64.99 | 43.15 | 0.72 | 90.78 | 81.88 | 0.94 |
| NAON-BART | 84.19 | 61.17 | 0.92 | 87.17 | 73.90 | 0.93 | 54.76 | 30.64 | 0.77 | 79.16 | 62.63 | 0.87 | 79.57 | 55.59 | 0.87 | 95.13 | 89.07 | 0.97 |

2019), **SE-Graph** encoding paragraph with sentence entities (Yin et al., 2019), and **En-Ptr** enhancing pointer network for sentence ordering with pairwise information (Yin et al., 2020) of generation based strategy. The performance comparisons of different compared methods are illustrated in Table 1. We compare our method with the baselines from several perspectives to analyze its performance comprehensively.

**General Comparison:** Standard performances of different methods, including our NAON, are shown in Black in the table. First, comparing with **generation-based** methods (*e.g.*, AON, En-Ptr, and TGCM), our method is clearly better as it greatly outperforms them on four datasets and is almost competitive on the NSF dataset. In particular, our NAON achieves best $\tau$ across all datasets, which suggests that it performs more like humans than other methods as the metric $\tau$ shows consistency of models with human judgments (Kumar et al., 2020). Second, compared with **ranking-based** methods, NAON outperforms most of them except B-TSort. However, B-TSort achieves slightly better performance than NAON mainly because of the relative order it takes advantage of, which also greatly increases computational costs.

**Relative Order Exploration:** Previous research has suggested that explicitly modeling the relative position of sentence pairs can bring considerably improve the sentence ordering models (Cui et al., 2020). Therefore, we further test our model with the boost of relative order (RO) modeling fol-

lowing (Cui et al., 2020), dubbed as NAON-RO. Furthermore, we report the performance of the existing method BERSON, which also applies RO modeling. First, it is clear that NAON-RO and BERSON gain significant improvement, showing the effectiveness of RO modeling. Second, our NAON-RO performs better than BERSON extensively in terms of all the datasets and metrics. It demonstrates that with the enhancement of RO modeling, the non-autoregressive nature of our method is still superior to the autoregressive for the sentence ordering task.

Despite of the superb of the RO-enhanced model in terms of the ordering accuracy, we take the plain model as the standard NAON considering the computational or memorial cost. The time and space complexity both turn into $O(N^2)$ with the RO modeling. For example, it costs 12GB or more memory to process a paragraph with more than 18 sentences and takes several days to train, and both costs will grow quadratically. Nevertheless, it still needs to mention that RO modeling is one of the potential directions to boost the sentence ordering models by exploring relative dependency. We hope in the future more computation-friendly algorithms would be developed to this end.

**Suitable Pre-training Task and Model:** Enhancing representations with effective pre-training process is nowadays a general way to improve the performance of models for higher-level tasks. RE-BART (Basu Roy Chowdhury et al., 2021) applies the BART as the pre-training model for sentence

Table 2: Accuracy comparison of predicting the first and last sentences on arXiv and SIND.

| Model | arXiv | | SIND | |
|---|---|---|---|---|
| | head | tail | head | tail |
| Pairwise | 84.85 | 62.37 | - | - |
| LSTM-Ptr | 90.47 | 66.49 | 74.66 | 53.30 |
| AON | 91.00 | 68.08 | 76.00 | 54.42 |
| SE-Graph | 92.28 | 70.45 | 78.12 | 56.68 |
| En-Ptr | 92.76 | 71.49 | 78.08 | 57.32 |
| TGCM | 92.46 | 69.45 | 78.98 | 56.24 |
| RankTxNet | 92.97 | 69.13 | 80.32 | 59.68 |
| NAON | **93.92** | **72.58** | **81.47** | **61.82** |
| BERSON | 94.75 | 76.69 | 84.95 | 64.87 |
| NAON-RO | 94.83 | 76.71 | 84.98 | 64.91 |
| RE-BART | 96.46 | 80.62 | 87.97 | 73.02 |
| NAON-BART | 98.98 | 85.91 | 90.10 | 80.99 |

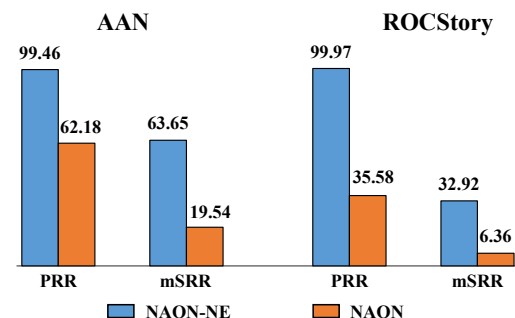

Figure 4: Repetition evaluation for NAON with and without exclusive loss, which do not apply the **selective and removing strategy** during inference. Smaller value indicates less repetition.

ordering, which already includes the *sentence permutation* task in the pre-training stage, and achieves excellent performance. Therefore, we also equip our non-autoregressive ordering strategy with the pre-trained BART model, namely NAON-BART, and examine its effectiveness. As the results shown in the last two blue rows in Table 1, empowering our NAON with BART pre-trained weights, it gains marvelous improvement and sets a new SOTA. This observation implies that: 1) our NAON could be applicable to various generation-based ordering models and boost their performance, and 2) exploring appropriate proxy tasks during pre-training might be beneficial for sentence ordering. Besides, these results also imply that our NAON is applicable to various generation-based models and can boost their performances by exploring the bilateral dependency of sentences.

### 3.2 Predicting Head and Tail

As mentioned in (Gong et al., 2016; Chen et al., 2016; Cui et al., 2018), the first and the last sentences play an important role in sentence ordering due to their special positions. Following previous works, we conduct the experiments on arXiv and SIND datasets and illustrate the comparison in Table 2. We can observe that our NAON outperforms all the baselines demonstrating superiority. Besides, the autoregressive fashion takes the previous output as current input, the first sentence prediction is much more crucial than others. While our non-autoregressive decoder is not subject to this limitation, the accuracy of the first sentence predic-

tion is still better than the last one. This implies that there may exist distinct signals in the first sentence, which indicates the important potential of exploiting such signals underlying sentences. Besides, the experiments of integrating NAON with BERSON and BART achieve similar and consistent results, further verifying the effectiveness of our NAON.

### 3.3 Diving into Exclusive Loss

As illustrated in the fifth and sixth lines from the bottom of Table 1, the proposed exclusive loss improves the sentence ordering performance significantly. Nevertheless, it is hard to investigate how it alleviates the repetition problem exactly. Towards this end, we produce the inference results before the *selective and removing* procedure, in which repetition exist. We conduct two levels of repetition evaluation. Specifically, the Paragraph Repetition Ratio (**PRR**) evaluates the paragraph-level repetition which equals 1 when a repetitive sentence exists in the paragraph otherwise 0. We also calculate the mean of the Sentence Repetition Ratio (**mSRR**) to evaluate from the sentence level by dividing the number of repetitive sentences by the total number of sentences in the paragraph.

We investigate the repetitive predictions from the NAON model *with and without exclusive loss* on AAN and ROCStory datasets. From the results exhibited in Figure 4, we can see that without exclusive loss, NAON is highly likely to generate paragraphs with repetitive sentences (PRR is 99.46% for AAN and 99.97% for ROCStory)[1]. While con-

---

[1]Note that our full model equipped with *greedy selective and removing strategy* will not generate repetition order. Here we remove this strategy to deeply probe and investigate the effectiveness of the **exclusive loss** for alleviating repetition issue from the training side.

strained by our novel exclusive loss during training, the model remarkably decreases the repetition ratio to 62.18% and 35.58%, which means that the exclusive constraint between sentences and positions makes great contributions in alleviating the repetition issue. At sentence level, the repetitive ratios for two datasets also drop dramatically with the introduction of exclusive loss, even achieving 6.36 indicating that there only exist a very small proportion of sentences repeated in the prediction. Based on the above observations, we can conclude that our exclusive loss can effectively alleviate the repetition problem of NAD, and thereby improve its performance for sentence ordering.

## 4 Accelerating Sentence Ordering

Non-autoregressive strategy implementing parallel decoding is with low latency and is more efficient than the autoregressive one (Gu et al., 2018). In this part, we investigate the time efficiency of our NAON empirically. We compare the inference time of NAON and AON, which share the same encoder and employ different decoders, *i.e.*, non-autoregressive and autoregressive decoders. Table 3 exhibits the inference time (measured by second) comparison between NAON and AON on each dataset, as well as the ratio comparing with NAON. We also include the comparison with B-TSort, the SotA ranking-based method, which shows huge time costs because it computes the relative orders of all the sentence pairs. From the above results, we observe that the proposed NAON gains significant speed-up on all datasets. Besides, since the average number of sentences per paragraph in NIPS, AAN, and NSF are larger than SIND and ROCStory, they benefit more from the parallel decoding of the non-autoregressive strategy. Such results are also consistent with our intuition that paragraph with more sentences takes more time with autoregressive methods.

## 5 Related Works

### 5.1 Sentence Ordering

Early works on sentence ordering attempt to utilize probabilistic transition method based on linguistic features (Lapata, 2003), content model (Barzilay and Lee, 2004) and entity-based approaches (Barzilay and Lapata, 2008; Prabhumoye et al., 2020). Recent neural models for the sentence ordering task are built on generation or ranking structure (Chen et al., 2016; Gong et al., 2016; Pandey

Table 3: Efficiency comparison of autoregressive and non-autoregressive strategies during inference, as well as B-TSort. We conduct this experiment on an NVIDIA TITAN V, and use **second** to measure the inference time of each dataset. **ratio** denotes the accelerating ratio comparing NAON to AON and B-TSort.

| Dataset | NAON | AON | | B-TSort | |
|---|---|---|---|---|---|
| | time | time | ratio | time | ratio |
| NIPS | 4.35 | 6.55 | 1.51× | 93.22 | 21.43× |
| AAN | 13.82 | 21.92 | 1.59× | 375.82 | 27.19× |
| NSF | 123.26 | 187.76 | 1.52× | 1848.91 | 15.00× |
| arXiv | 1040.81 | 1401.88 | 1.35× | - | - |
| SIND | 42.47 | 54.53 | 1.28× | 583.40 | 13.74× |
| ROCStory | 45.98 | 59.64 | 1.30× | - | - |

and Chowdary, 2020; Logeswaran et al., 2018; Yin et al., 2020). For ranking-based strategy, Chen et al. (2016) introduced a pairwise model which ranks relative order for sentence pairs. Kumar et al. (2020) employed BERT-improved feedforward neural networks to generate scores for each sentence and optimized the scores with several ranking losses, *e.g.*, pairwise ranking loss. Prabhumoye et al. (2020) proposed to constrain the relative order between paired sentences and introduced the classic topological sort algorithm to find the gold order, which achieves considerable improvement.

For generation-based strategy, Gong et al. (2016) firstly utilized an RNN to encode sentences and Logeswaran et al. (2018) explored LSTM-based attention encoder, both works employed a similar recurrent pointer net to generate the coherent paragraph sentence-by-sentence. However, since the input of sentence ordering is a set of unordered sentences, it is inappropriate to model the unordered sentences in an autoregressive manner. To address this issue, Cui et al. (2018) proposed to use Transformer but without positional embedding to encode the unordered sentences. Most of the follow-up methods also applied the order invariant Transformer as paragraph encoder (Yin et al., 2019, 2020; Kumar et al., 2020; Cui et al., 2020). Besides, other attempts also have been explored for order invariant encoding. Yin et al. (2019) devised a graph structure to encode the input order-free sentence set. Wang and Wan (2019) proposed a hierarchical attention network composed of a word encoder and a sentence encoder-decoder. Cui et al. (2020) and Li et al. (2022) further took advantage of BERT to explore the deep semantic connection and relative order between sentences. Basu Roy Chowdhury et al. (2021) employed the BART as backbone, which benefited from the sentence permutation task during pre-training and set a strong SotA.

## 5.2 Non-Autoregressive Transformer

Non-autoregressive transformer (NAT) (Gu et al., 2018) has been proposed as an effective non-autoregressive decoding method in machine translation. NAT predicts the output sequence based on source sequence and latent representation via parallel fashion. In order to narrow the performance gap between NAT and autoregressive transformer (AT), various approaches have been proposed in many research areas (Guo et al., 2020; Yang et al., 2021; Bin et al., 2023). Several methods (Lee et al., 2018; Guo et al., 2019; Wang et al., 2019; Li et al., 2019; Guo et al., 2020) proposed to modify the latent presentation by making use of extra refinement process, designing auxiliary modules and latent forms. Another line of research (Ghazvininejad et al., 2019; Stern et al., 2019; Gu et al., 2019) tended to achieve a trade-off between performance and inference efficiency by using semi-NAT architecture. Besides, many researchers had made attempts to introduce NAT into different sequence generation tasks like speech recognition (Ren et al., 2019) and dialog state tracking (Le et al., 2020). Moreover, it is worth noting that many NAT related pre-trained generation models (Chan et al., 2019; Qi et al., 2020) had been proposed to improve the generation results of pre-trained models. Our work makes the first attempt to accomplish sentence ordering task by a NAT style model.

In order to perform parallel decoding, sequence length should be determined firstly in NAT. The pioneer work in NAT (Gu et al., 2018) designed fertility predictor to predict the copy times of each token in the source sequence. By summarizing the copy times of all the tokens, model can obtain the target length of output sequence. Later, (Lee et al., 2018) proposed to use a separate length predictor to obtain the target length for the inference process. Some semi-NAT models (Stern et al., 2019; Gu et al., 2019) even proposed to implement insertion or deletion-based methods to control the length of generated sequences. Ran et al. (2021) introduced an intermediate pseudo-translation to align the order between source and target languages, which could help model the conditional dependencies of the target words and encourage the decoder to choose right words. Du et al. (2021) claimed the penalty for word order errors should not be included, because the semantic alignment between source and target words could be order-agnostic. In the ordering problem (Bin et al., 2022), as the number of sentences is fixed in the input, our NAON model does not need to consider this issue and can fully explore the potential of NAT in this task.

NAT models usually suffer from repetitive generation issue due to conditional independence between output representations. For machine translation, Gu et al. (2018) proposed to use fertility *i.e.*, copy times of each source token, to bridge the encoding and decoding processes, copying input tokens for several times as inputs of the decoder. Due to the conditional independence between output tokens, NAT model tends to generate repeated tokens in the output sequence. To alleviate this issue, Wang et al. (2019) and Li et al. (2019) introduced auxiliary loss functions to deal with the problem of repetition. Recently, semi-NAT style models (Stern et al., 2019; Gu et al., 2019) proposed new generation approaches to alleviate repetition issue. In our NAON model, we introduce an exclusive loss to tackle this problem.

## 6 Conclusion

We presented a novel non-autoregressive ordering network for sentence ordering, which consists of a basic sentence encoder, a contextual sentence encoder, and a non-autoregressive decoder. Different from existing autoregressive methods, the proposed NAON predicted sentence for each position in parallel, which also leveraged bilateral dependencies among sentences, and demonstrated competitive performance. To alleviate the repetition problem of the proposed NAON, we devised an exclusive loss to constrain the exclusiveness matching between sentences and positions. The experimental results on several common-used datasets demonstrated the effectiveness of the proposed approach. The analyses of exclusive loss also indicate that alleviating repetition issue could improve the generation performance of the non-autoregressive decoder.

## 7 Acknowledgement

This research is partially supported by the National Natural Science Foundation of China under grant 62102070, 62220106008, and U20B2063, and partially supported by Sichuan Science and Technology Program under grant 2023NSFSC1392. This research is supported by A*STAR, CISCO Systems (USA) Pte. Ltd and National University of Singapore under its Cisco-NUS Accelerated Digital Economy Corporate Laboratory (Award I21001E0002).

## 8 Limitations

In this work, we propose to employ non-autoregressive Transformer (NAT) as the decoder for sentence ordering, which could leverage bilateral dependencies between sentences, different with previous autoregressive decoders (*e.g.*, RNN and LSTM) which explore only unilateral dependencies. However, to clearly analyze the particular priority of the NAT structure for sentence ordering, we adopt the vanilla NAT and simple repetition mitigation strategy for our NAON, ignoring further improvements or advances on NAT. Therefore, only use the vanilla NAT could be a limitation of our work. Despite that previous work or experimental results have suggested techniques such as exploring relative order exploration or applying the BART model could further improve the performance of sentence ordering, we do not want these detailed techniques to overshadow the main deployment of non-autoregressive decoding. We hope our work could bring some insights to the research of ordering problem and encourage more attempts at integrating advanced techniques with NAT.

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

## A  Sentence Ordering with ChatGPT

With the blooming of Large Language Models (LLMs), especially the great success of GPT series, *e.g.*, ChatGPT[2] and GPT-4[3], many NLP tasks have been proposed to integrate with LLMs based on suitable prompts and demonstrated marvelous results and performance. We therefore take an attempt to recover the coherence of sentences with ChatGPT API. We test ChatGPT on NIPS and SIND datasets only, due to the expensive costs of ChatGPT API calling, and the results are shown in Table 4. Before the test, we expected ChatGPT could achieve marvelous ordering performance, *e.g.*, more than 95% in Acc, but the results show that it seemed to be not skilled in sentence ordering problems, and shocked us with dramatically low

---

[2]https://openai.com/blog/chatgpt
[3]https://openai.com/gpt-4

| Prompt | NIPS | | | SIND | | |
|---|---|---|---|---|---|---|
| | Acc | PMR | $\tau$ | Acc | PMR | $\tau$ |
| **List** | | | | | | |
| -P0 | 24.87 | 6.12 | 0.26 | - | - | - |
| -P1 | 27.40 | 6.12 | - | - | - | - |
| -P2 | 42.44 | 17.82 | 0.39 | 37.32 | 9.95 | 0.27 |
| -P3 | 46.89 | 19.41 | 0.5 | 41.34 | 13.1 | 0.45 |
| **Dict** | | | | | | |
| -P1 | 30.48 | 6.12 | 0.39 | - | - | - |
| -P2 | 56.92 | 24.93 | 0.7 | 44.96 | 17.07 | 0.49 |
| -P3 | 58.10 | 28.8 | 0.71 | 52.38 | 22.71 | 0.57 |
| TGCM | 59.43 | 31.44 | 0.75 | 38.71 | 15.18 | 0.53 |
| B-TSort | 61.48 | 32.59 | 0.81 | 52.23 | 20.32 | 0.60 |
| NAON | 64.43 | 33.02 | 0.80 | 52.38 | 19.13 | 0.60 |

Table 4: The test performance using ChatGPT API on NIPS and SIND datasets. **List** and **Dict** mean that the unordered sentences are input with the format of list and dict. P* indicate different prompts that could be found in Figure 6. To reduce the token costs of API calling, we only test on NIPS (only 376 samples) for exploratory study for inferior prompts at early stage, e.g., P0 and P1. We also list three ordering models for reference.

performance. We analyze the results and assume the reasons may come from two points. The first one is that, our prompts are not very well to instruct the ChatGPT to effectively recover the coherence and reorder the sentences. Because we have also tried to revise our prompts several times to make the prompt more clear and more instructive (in Figure 6), the performance could be significantly improved with better prompts. Therefore, if we can find more suitable and instructive prompts to encourage the ChatGPT to understand the semantics better, it should work well for recovering the coherence of a set of unordered sentences. We will keep taking further attempts at this point. The second one might come from the model side of ChatGPT. As we know, GPT series are decoder-only and designed for generative pre-training and exhibit extraordinary abilities of generation, e.g., creative writing, and story completion, etc. While sentence ordering is a semantic coherence understanding task, where the output is strictly constrained by the input. We therefore suppose that the strict output may limit the ability of GPT model for sentence ordering.

**Analyses of Input Styles:** For the unordered sentences, we have tried two ways to feed them to ChatGPT: List and Dict, detailed in Figure 5. The performance comparisons are in Table 4, from which we can observe that ChatGPT works better with the "Dict" format than "List". We also have further investigated the reasons, and found Chat-

GPT may fail to count the number of sentences with "List" input. For example, it might output four or six indices to indicate the new order of five sentences (A true output case is shown in Figure 7). The failure cases are about 5% and 6.8% for NIPS and SIND datasets. While the "Dict" input provides explicit labels, e.g., s1 and s2, for each sentence, and significantly avoids such outputs (decreased to 0.8% and 0.5% for NIPS and SIND).

**Analyses of Prompts:** As we know, the quality of prompts could introduce significant impacts and result in violent fluctuations of performance. During our experiments, we have tried to make our prompts clearer and more suitable for sentence ordering, shown in Figure 6. We note that only request ChatGPT to output the order without the coherent paragraph (e.g., Prompt-0 and Prompt-1), it works extremely badly (P0 and P1 in Table 4). When we instruct it first to output the coherent paragraph and give the new order in subsequent (Prompt-2), it gains significant improvement. This is also consistent with that LLMs are somehow heuristic and could be guided with step-by-step reasoning (Wei et al.). We observe it might predict the number of indices or sentence signs inconsistent with the true number of sentences for both List and Dict inputs. We further explicitly indicate the number of sentences in the prompt (Prompt-3), the inconsistent number of sentences could be effectively avoided, resulting in 1/367 (0.27%) and 3/5055 (0.06%) for NIPS and SIND datasets. Though the clearer prompts significantly boost the sentence ordering of ChatGPT, it is still far from our expectation, even compared to the relatively small pre-trained models, e.g., BERT and BART. Therefore, we will keep designing more instructive prompts for sentence ordering in future works.

For this part of work, we would like to thank Andrew Ng and Isa Fulford for providing the prompt instructions in an open-source course[4]. We also thank all the people sharing tricks online for prompt engineering.

## B    Details of Experiments

### B.1    Datasets

As aforementioned, to evaluate the effectiveness of the proposed NAON, we conduct experiments on several commonly used datasets for comprehensive analyses, including four datasets collecting the

---

[4]https://www.deeplearning.ai/short-courses/chatgpt-prompt-engineering-for-developers/

abstract of academic papers and two story understanding corpora.

- **AAN** (Wang et al., 2018): the abstracts of papers published on ACL Anthology Network until 2016. We use the data provided by (Logeswaran et al., 2018), as well as the data split.

- **NIPS**: the paper abstracts of NIPS conference in years 2005-2013/2014/2015 for training/validation/testing, provided by (Logeswaran et al., 2018).

- **NSF**: the abstracts of NSF Research Award, also provided by (Logeswaran et al., 2018).

- **arXiv** (Chen et al., 2016): contains abstracts of seven disciplines, including *physics, computer science* and *etc*, on arXiv website.

- **SIND** (Huang et al., 2016): stories for images in sequence, collected for visual storytelling. Each story consists of 5 sentences.

- **ROCStory** (Mostafazadeh et al., 2016): originally collected for commonsense story understanding, which provides 5 sentences in each story.

## B.2 Evaluation Metrics

In this work, we employed three metrics, including **Acc**, **PMR**, and $\tau$, to evaluate the performance and compare it with the baselines. The details of the metrics are as follows:

- **Perfect Match Ratio (PMR)** calculates the exact matching on paragraph level, which is the most rigid one of existing metrics.

- **Accuracy (Acc)** takes a more relaxed measure strategy by calculating the accuracy of absolute position prediction on sentence level. Obviously, it may fail to evaluate the coherence of relative order, *e.g.*, scoring 0 for $[2, 3, 4, 1]$ with the ground-truth $[1, 2, 3, 4]$.

- **Kendall's tau ($\tau$)** measures the relative order between all sentence pairs in a predicted paragraph, which is defined as:

$$\tau = 1 - 2 * I / \binom{N}{2},$$

where $I$ is the number of inversion between two sentences comparing with gold order, and

N is the total number of sentences in the paragraph.

All the metrics indicate the better performance by higher score.

## B.3 Experimental Settings

We implement our model using PyTorch, and employ the Adam algorithm (Kingma and Ba, 2015) to optimize all the models. The initial learning rate is set as $lr_i = 1e^{-4}$ for the contextual encoder and non-autoregressive decoder, and $lr_b = 5e^{-5}$ for fine-tuning BERT-BASE with 12 Transformer blocks. The training process is terminated if the performance of validation worse than the best one for 5 times. To prevent over-fitting, we adopt weight decay and dropout operation during training, the parameters of which are $\lambda = 1e^{-5}$ and $p = 0.1$. The hidden size of transformer blocks are 1024 and 2 multi-head inter-attention blocks in decoder. The number of Transformer blocks in encoder are specific to different datasets, in particular, 2 for SIND, ROCStory and NIPS, 4 for NSF and arXiv, 6 for AAN. All the experiments are conducted on a workstation with 8 NVIDIA Titan V GPUs.

**List Input:**

["some of these crafts are very unique and take a lot of talent to make. ", " some of the crafters even dress up in unique costumes as part of their selling act.", " lots of folks come out and set up tables to sell their crafts. ", "the local parish holds a craft show each year. ", " folks of all ages come out to peruse the crafts for sale. "]

**Expected Output:**

**Output1:** The coherent order is: [4, 3, 1, 5, 2]   *--(for the index only output)*
**Output2:** The coherent paragraph is: the local parish holds a craft show each year. lots of folks come out and set up tables to sell their crafts. some of these crafts are very unique and take a lot of talent to make. folks of all ages come out to peruse the crafts for sale. some of the crafters even dress up in unique costumes as part of their selling act. Therefore, the coherent order is: [4, 3, 1, 5, 2]   *--(for the coherent paragraph and index output)*

**Other Outputs of ChatGPT:**

**Output3:** The coherent order is: [3, 2, 0, 4, 1]
**Output4:** The coherent order is: [4][3][1][5][2]
**Output5:** The coherent order is: [3][2][0][4][1]
**Output6:** The coherent order is: ["the local parish holds a craft show each year. ", " lots of folks come out and set up tables to sell their crafts. ", " some of these crafts are very unique and take a lot of talent to make. ", " folks of all ages come out to peruse the crafts for sale. ", " some of the crafters even dress up in unique costumes as part of their selling act."]

---

**Dict Input:**

{"s1": " some of these crafts are very unique and take a lot of talent to make. ", "s2": " some of the crafters even dress up in unique costumes as part of their selling act.", "s3": "lots of folks come out and set up tables to sell their crafts. ", "s4": "the local parish holds a craft show each year.", "s5": "folks of all ages come out to peruse the crafts for sale. "}

**Expected Output:**

**Output1:** The coherent order is: [s4, s3, s1, s5, s2]   *--(for the index only output)*
**Output2:** The coherent paragraph is: the local parish holds a craft show each year. lots of folks come out and set up tables to sell their crafts. some of these crafts are very unique and take a lot of talent to make. folks of all ages come out to peruse the crafts for sale. some of the crafters even dress up in unique costumes as part of their selling act. Therefore, the coherent order is: [s4, s3, s1, s5, s2]   *--(for the coherent paragraph and index output)*

**Other Outputs of ChatGPT:**

**Output3:** The coherent order is: [s4][s3][s1][s5][s2]
**Output4:** The coherent order is: ["the local parish holds a craft show each year. ", " lots of folks come out and set up tables to sell their crafts. ", " some of these crafts are very unique and take a lot of talent to make. ", " folks of all ages come out to peruse the crafts for sale. ", " some of the crafters even dress up in unique costumes as part of their selling act."]

Figure 5: The inputs and outputs for ChatGPT API calling. We tested two formats of input: List and Dict, and the corresponding expected outputs are the indices and keys of sentences. We note that ChatGPT does not always give the outputs as we expected, it also outputs some cases with other styles, as shown in the "other outputs of ChatGPT" blocks. When we make the output more formatted, *e.g.*, the "output2", it could avoid the unexpected outputs.

**Prompt-0:**
Please analyze the following list of unordered sentences delimited by triple backticks and determine the coherent order.
Once you have identified the correct order, please output the INDEX in the following format: [0, 2, 1, 4, 3].
Please DON'T provide any additional outputs or information other than the requested INDEX.
The unordered sentences:
```{}```

**Prompt-1:**
From the point of semantic coherence, please REORDER the following sentences, delimited by triple backticks, to a coherent paragraph.
Once you have recovered the coherence, please output the NEW ORDER in the following format: The coherent order is: [s1, s3, s2, s5, s4]. **([1, 3, 2, 5, 4] for LIST input)**
Please be CONCISE and DO NOT provide any additional outputs or information other than the requested INDEX.
The sentences are:
```{}```

**Prompt-2:**
Please REORDER the following sentences, delimited by triple backticks, to a coherent paragraph.
The sentences are:
```{}```
Once you have recovered the coherence, please output in the following format: ```The coherent paragraph is: COHERENT PARAGRAPH. Therefore, the coherent order is: [SIGNS OF SENTENCES]. **([INDICES OF SENTENCES] for LIST input)**```
Please be CONCISE and DO NOT provide any additional outputs or information other than the requested output.

**Prompt-3:**
Please REORDER the following {} sentences, delimited by triple backticks, to a coherent paragraph.
The sentences are:
```{}```
Once you have recovered the coherence, please output in the following format: ```The coherent paragraph is: COHERENT PARAGRAPH. Therefore, the coherent order is: [SIGNS OF SENTENCES]. **([INDICES OF SENTENCES] for LIST input)**```
Please be CONCISE and DO NOT provide any additional outputs or information other than the requested output.

Figure 6: The prompt templates used in our ChatGPT API test. For "Prompt-0", we have only tested LIST input, and tested LIST and DICT inputs for other prompts. The contents in brackets of "Prompt-1/2/3" are not a part of the corresponding prompt, which are used to replace the output format for different input formats. The corresponding formats of input and output could be found in Figure 5.

**Input and Prompt:**
Please REORDER the following sentences, delimited by triple backticks, to a coherent paragraph.
The sentences are:
[" we also met lots of locals who were quirky and fun. ", " we decided that the next time we visited we would pay more attention to the history of thailand. ", " we were not sure what to expect from our trip to thailand. ", " we found a lot of beautiful architecture and art and a lot of it was religious. ", " we explored a bit for ourselves and found some areas where there was a lot of poverty. "]
Once you have recovered the coherence, please output in the following format: ```The coherent paragraph is: COHERENT PARAGRAPH. Therefore, the coherent order is: **[INDICES OF SENTENCES].** Please be CONCISE and DO NOT provide any additional outputs or information other than the requested output.

**Number of predicted sentences increases:**
**Output:** The coherent paragraph is: we were not sure what to expect from our trip to thailand. we found a lot of beautiful architecture and art. a lot of it was religious. we explored a bit for ourselves and found some areas where there was a lot of poverty. we also met lots of locals who were quirky and fun. we decided that the next time we visited we would pay more attention to the history of thailand.
Therefore, the coherent order is: [3, 4, 1, 5, 0, 2]
**Ground Truth Order:** [3, 5, 1, 4, 2]

**Input and Prompt:**
Please REORDER the following sentences, delimited by triple backticks, to a coherent paragraph.
The sentences are:
[" some of the guests will be sleeping over tonight and sleeping on the futons in the living room. ", " we finally see the exit sign on the highway for the party. ", " when all the guests have gone home, the host is vacuuming the living room. ", " there is a long drive to the party tonight. ", " when we got there, all our friends are already there. "]
Once you have recovered the coherence, please output in the following format: ```The coherent paragraph is: COHERENT PARAGRAPH. Therefore, the coherent order is: **[INDICES OF SENTENCES].** Please be CONCISE and DO NOT provide any additional outputs or information other than the requested output.

**Number of predicted sentences is reduced:**
**Output:** The coherent paragraph is: there is a long drive to the party tonight. when we got there, all our friends are already there. some of the guests will be sleeping over tonight and sleeping on the futons in the living room. when all the guests have gone home, the host is vacuuming the living room. Therefore, the coherent order is: [4, 5, 1, 3]
**Ground Truth Order:** [4, 2, 5, 1, 3]

Figure 7: The cases of inconsistent number of sentences predicted by ChatGPT. The top one predicts one more index for the sentences, while the bottom one discards one sentence.

**Input and Prompt:**
Please REORDER the following sentences, delimited by triple backticks, to a coherent paragraph.
The sentences are:
[" there are lots of rides to enjoy this year. ", " the family takes a trip to the local carnival. ", " there are also lots of games and prizes to win. ", " there are even rides for folks as young as this small boy. ", " although some of the games seem fixed and a waste of money. "]
Once you have recovered the coherence, please output in the following format: ```The coherent paragraph is: COHERENT PARAGRAPH. Therefore, the coherent order is: **[INDICES OF SENTENCES].** Please be CONCISE and DO NOT provide any additional outputs or information other than the requested output.

**Output is Correct:**
**Output:** The coherent paragraph is: the family takes a trip to the local carnival. there are lots of rides to enjoy this year. there are even rides for folks as young as this small boy. there are also lots of games and prizes to win. although some of the games seem fixed and waste money.
Therefore, the coherent order is: [2, 1, 4, 3, 5]
**Ground Truth Order:** [2, 1, 4, 3, 5]

**Input and Prompt:**
Please REORDER the following sentences, delimited by triple backticks, to a coherent paragraph.
The sentences are:
[" every summer we head out west to texas to visit relatives and experience country life. ", " these are my cousins having a fun moment with each other. ", " and these are my friends and relatives while we check out the produce of their farm. ", " while it was hot and humid, we really enjoyed the country view with less noise and traffic and no buildings. ", " here we see my cousin's pigs and are they huge and noisy pigs. "]
Once you have recovered the coherence, please output in the following format: ```The coherent paragraph is: COHERENT PARAGRAPH. Therefore, the coherent order is: **[INDICES OF SENTENCES].** Please be CONCISE and DO NOT provide any additional outputs or information other than the requested output.

**Output is Wrong:**
**Output:** The coherent paragraph is: every summer we head out west to texas to visit my relatives and experience country life. while it was hot and humid, we really enjoyed the country view with less noise and traffic and no buildings. here we see my cousin's pigs and are they huge and noisy pigs. these are my friends and relatives while we check out the produce of their farm. these are my cousins having a fun moment with each other. Therefore, the coherent order is: [1, 4, 5, 3, 2]
**Ground Truth Order:** [1, 5, 2, 4, 3]

Figure 8: Two cases of SIND outputted by ChatGPT, inputting as "List" format with Prompt-3.