# OpenReview forum: "Non-Autoregressive Sentence Ordering"
_EMNLP/2023/Conference — EMNLP 2023 Findings_

### Official Review · Reviewer_FhyS · 2023-07-30

**Soundness:** 4

**Excitement:**

4: Strong: This paper deepens the understanding of some phenomenon or lowers the barriers to an existing research direction.

**Paper Topic And Main Contributions:**

This paper presents a new approach to solve the sentence ordering task with non-autoregressive methods. The method uses an order-invariant encoder to get sentence representations and then non-autoregressively predicts the order for each position using pointer networks. To avoid repetitions in non-autoregressive generation, the paper introduces an exclusive loss and a greedy selective and removing strategy. The experiments demonstrate that this proposed method outperforms previous generation methods and most ranking-based methods, while also being slightly faster than autoregressive generation methods.

**Questions For The Authors:**

* Eq(2): Should it be $o_1$ to $o_j$ instead of $s_1$ to $s_j$? (Where S is the set of $s_1$ to $s_N$)
* Eq(3): This is a similar problem. Additionally, since the non-autoregressive model predicts each token independently, the condition may not need to include $o_1$ to $o_N$.
* L536-L539: I couldn't find the speed comparison with B-Tsort.
* Previous research on non-autoregressive translation models generally believes that NAT models face challenges in capturing the correct order of predicted tokens. For instance, [1] proposes the use of an autoregressive model to guide the token generation order, while [2] introduces an order-agnostic loss to address the issue of NAT models mixing sentences with different orders. It would be helpful if the authors could discuss the distinctions between their findings and the earlier beliefs.

[1] Qiu Ran, Yankai Lin, Peng Li, Jie Zhou. Guiding Non-Autoregressive Neural Machine Translation Decoding with Reordering Information. AAAI 2021.
[2] Cunxiao Du, Zhaopeng Tu, Jing Jiang. Order-Agnostic Cross Entropy for Non-Autoregressive Machine Translation. ICML 2021.

**Reasons To Accept:**

* It's surprising that the proposed non-autoregressive model performs better than autoregressive models in sentence ordering tasks, considering non-autoregressive models are generally weak at modeling dependencies between generated tokens. This finding will be valuable for non-autoregressive model research.
* The experiments are solid. The comparison against chatGPT in the appendix is particularly interesting.

**Reasons To Reject:**

* Nothing particular.

**Reproducibility:**

4: Could mostly reproduce the results, but there may be some variation because of sample variance or minor variations in their interpretation of the protocol or method.

**Reviewer Confidence:**

4: Quite sure. I tried to check the important points carefully. It's unlikely, though conceivable, that I missed something that should affect my ratings.

---

> ### Author Rebuttal · Authors · 2023-08-28
>
> We thank and appreciate the reviewer's careful and valuable comments. To make our paper clearer, below we try to answer the questions and issues raised by the reviewer. Hope it could address the concerns of the reviewer. Thanks very much.
>
> **Quetion_1:** Eq(2): Should it be $o_1$ to $o_j$ instead of $s_1$ to $s_j$? (Where $S$ is the set of $s_1$ to $s_n$)
>
> **Answer_1:** Thanks for the review. The notations are right here. We apologize for the ambiguous description. The generation-based models **do not directly generate the gold order** of the sentences, they instead generate the coherent paragraph **sentence-by-sentence** to achieve the goal of recovering the cohrence of unordered sentences. Therefore, the output of each step is a coherent representation of each sentence, rather than the order, and the model tries to maximize the similarity between such coherent representation and previous context representation. This is similar with text generation,  which can classify the output embedding to words or maximize the similarity between output word embedding and input word embedding.
> In the revision, we will make the description in L170-172 clearer as follows:
> ```Existing autoregressive approaches generate the coherent paragraph sentence-by-sentence to recover the gold order, and optimize the model by maximizing:```
>
> $ \sum_{j=1}^{N}\log P(s_{j}|s_1, s_2, ...,s_{j-1}, S) $
>
> **Question_2:** Eq(3): This is a similar problem. Additionally, since the non-autoregressive model predicts each token independently, the condition may not need to include $o_1$ to $o_n$.
>
> **Answer_2:** Thanks for the suggestion. For $o_1$ to $o_n$, the condition does not need to include $o_1$ to $o_n$. Here we just want to emphasize the interaction between $o_1$ to $o_n$ during decoding. We will revise this in the revision, and emphasize the interaction with description.
>
>
> **Question_3:** L536-L539: I couldn't find the speed comparison with B-Tsort.
>
> **Answer_3:** We apologize for our carelessness during the space adjustment. Because our NAON employs very similar architecture with AON, except for the non-autoregressive decoder, comparing the speed of AON and NAON is the most proper setting to evaluate the efficiency of non-autoregressive decoding. We include B-TSort for additional reference to compare with other methods, but the space is limited. To meet the requirement of page limitation, we removed the results from Table 3 and adjusted it from two-column to single-column, but we forgot to remove corresponding descriptions in the analysis. We apologize for this carelessness. The inference time and acceleration ratio of **B-TSort** are listed below:
>
> \begin{matrix}
> \text{Dataset} & \text{B-TSort} & \text{NAON} & \text{Accelerating Ratio} \\\\ \hline
> \text{NIPS}       & 93.22   & 4.35   & 21.43  \times    \\\\
> \text{AAN}         & 375.82  & 13.82 & 27.19  \times   \\\\
> \text{NSF}        & 1848.91 & 123.26 & 15.00  \times  \\\\
> \text{arXiv}      & - & 1040.81 & -  \\\\
> \text{SIND}       & 583.40  & 42.47  & 13.74  \times  \\\\
> \text{ROCStory}   & -  & 45.98  & - \\\\ \hline
> \end{matrix}
>
> Because the original paper and released codes did not conduct experiments on arXiv and ROCStory datasets, we also did not test their method on these two datasets. We will include these results in the revision.
>
> **Question_4:** Previous research on non-autoregressive translation models generally believes that NAT models face challenges in capturing the correct order of predicted tokens. For instance, [1] proposes the use of an autoregressive model to guide the token generation order, while [2] introduces an order-agnostic loss to address the issue of NAT models mixing sentences with different orders. It would be helpful if the authors could discuss the distinctions between their findings and the earlier beliefs.
>
> **Answer_4:** Thanks for providing the literatures. Both works are aiming at the multimodal problem in machine translation with NAT and trying to tackle this problem from the perspective of generation order. ReorderNAT [1] introduces an intermediate pseudo-translation to align the order between source and target languages, which could help model the conditional dependencies of the target words and encourage the decoder to choose right words. OAXE [2] claims the penalty for word order errors should not be included, because the semantic alignment between source and target words could be order-agnostic. We agree with that NAT models face challenges in capturing the correct order of predicted tokens, but we claim that the order should be **‘absolute order’**. In other words, **it is hard to align a certain word to an absolute position.** Through the bidirectional interactions of attention, NAT could capture the **‘relative order’** dependencies between words/sentences. In sentence ordering problem, the goal is to capture the ‘relative order’ between sentences, therefore, we believe it could avoid the ‘absolute order’ challenge in NAT. Thanks again for the review, we will include the discussion and distinctions between the mentioned works [1, 2] and our NAON in the revision.

---

### Official Review · Reviewer_8B3p · 2023-08-04

**Soundness:** 3

**Excitement:**

3: Ambivalent: It has merits (e.g., it reports state-of-the-art results, the idea is nice), but there are key weaknesses (e.g., it describes incremental work), and it can significantly benefit from another round of revision. However, I won't object to accepting it if my co-reviewers champion it.

**Paper Topic And Main Contributions:**

This paper proposes NAON, a novel non-autoregressive approach for sentence ordering. The key contribution is using a non-autoregressive transformer decoder to predict sentences in parallel rather than sequentially. Non-autoregressive decoding could explore bilateral dependencies between sentences and enable parallel prediction during inference. Furthermore, the authors introduce an exclusive loss to alleviate repetition issues in non-autoregressive decoding.

Overall the paper is well-written and easy to follow. The experiments on several benchmark datasets demonstrate the effectiveness of the proposed NAON model.

**Questions For The Authors:**

- Question A: The non-autoregressive decoding seems to use a basic greedy selection approach. Have you explored integrating more advanced NAT techniques like iterative refinement? How much gain can these provide?

- Question B: You test integrating NAON with BART, but only report accuracy results. Can you also analyze the impact on efficiency and repetitions? How does BART integration compare to your exclusive loss?

- Question C: The ChatGPT experiments are interesting but limited. Can you provide more details on the prompt engineering process and lessons learned on how to prompt sentence order effectively?


**Reasons To Accept:**

- Proposes a novel application of non-autoregressive decoding for sentence ordering, which is interesting to the NLP community

- Well-designed model with contextual encoders and NAT decoder that effectively exploits bilateral dependencies

- Detailed analyses like quantifying repetition reduction from the exclusive loss

- Explores integrating NAON with pre-trained models to push the state-of-the-art further

Overall, this work demonstrates a novel application of an important technique (NAT) on an important problem (ordering) with empirical. Accepting this could significantly broaden the scope and advance the state-of-the-art of non-autoregressive decoding for sequence generation tasks.

**Reasons To Reject:**

- The non-autoregressive decoding approach is basic without incorporating advanced NAT techniques like iterative refinement. This risks presenting a simple application without pushing boundaries.
- The ChatGPT experiments, while interesting, seem preliminary and exploratory. Relying heavily on these incomplete experiments may be risky.
- There could be more ablation studies to analyze the impact of various design choices instead of just the loss.

**Reproducibility:**

3: Could reproduce the results with some difficulty. The settings of parameters are underspecified or subjectively determined; the training/evaluation data are not widely available.

**Reviewer Confidence:**

4: Quite sure. I tried to check the important points carefully. It's unlikely, though conceivable, that I missed something that should affect my ratings.

---

> ### Author Rebuttal · Authors · 2023-08-28
>
> We thank and appreciate the reviewer's careful and valuable comments. To make our paper clearer, below we try to answer the questions and issues raised by the reviewer. Hope it could address the concerns of the reviewer. Thanks very much.
>
> **Question_1:** The non-autoregressive decoding seems to use a basic greedy selection approach. Have you explored integrating more advanced NAT techniques like iterative refinement? How much gain can these provide?
>
> **Answer_1:** Thanks for the suggestive reviews. Our NAON indeed employs vanilla non-autoregressive decoding and a basic greedy selection strategy, and does not integrate with other advanced techniques, e.g., iterative refinement, to further improve the performance. The reason is that all the existing generation-based methods employ autoregressive decoding strategy, including RNN-based and Transformer, for sentence ordering. Our **NAON is the first one** to employ non-autoregressive decoding to explore bilateral dependencies for sentence ordering. To evaluate the performance of non-autoregressive decoding fairly and purely, we only employ vanilla NAT without any advanced techniques. We also verify the generalization ability of our NAON by integrating with BERSON and BART, which means our NAON could be further improved with advanced techniques. As Reviewer 7JW7 mentioned, ```RE-BART is hard to beat and held the SOTA for nearly 2 years```. **When equipped with our non-autoregressive decoder, NAON-BART substantially outperforms RE-BART by a large margin (shown in Table 1, averaging relative improvements across six datasets w.r.t Acc: 9.17%, PMR: 8.13%, and $\tau$: 5.35%). Therefore, from this perspective, we believe we have pushed the boundaries of sentence ordering.** Though we have not explored more techniques in this work, as mentioned in **Section Limitation**, we hope our work, the first attempt at non-autoregressive decoding, could bring some insights to the research of ordering problem and encourage more attempts at integrating advanced techniques to facilitate non-autoregressive decoding. Thanks for the constructive review.
>
>
> **Question_2:** You test integrating NAON with BART, but only report accuracy results. Can you also analyze the impact on efficiency and repetitions? How does BART integration compare to your exclusive loss?
>
> **Answer_2:** Thanks for the suggestive comments. As the reviewer suggested, we test the efficiency of RE-BART and NAON-BART on ROCStory, because the authors of RE-BART only released the checkpoint on this dataset (We will try to train and test RE-BART on other datasets, and include the results in the revision). The results are as follows:
>
> \begin{matrix}
> \hline
> \text{Dataset}    &  \text{RE-BART} & \text{NAON-BART} &   \text{Accelerating Ratio}          \\\\  \hline
> \text{ROCStory}       & 851.59   & 524.32   & 1.62\times  \\\\ \hline
> \end{matrix}
>
> From the results we can see, replacing the autoregressive decoder in RE-BART with our non-autoregressive decoder could gain 1.62 times acceleration, which shows similar performance in Table 3.
> For the repetition performance of NAON-BART, the results are as:
>
> \begin{matrix}
> \hline
> \text{Dataset}    &  \text{PRR} & \text{mSRR}           \\\\ \hline
> \text{ANN}       & 52.08   & 13.05     \\\\
> \text{ROCStory}       & 22.67   & 3.39     \\\\ \hline
> \end{matrix}
>
> Comparing with the results shown in Figure 4, we observe significantly decrease of repetition comparing with vanilla NAON, which means the integration with BART also further improve the overall performance of sentence ordering.
>
> For the question “How does BART integration compare to your exclusive loss”, BART and exclusive loss are two ways to reinforce our simple and naïve NAON, but **they contribute from two different aspects**. Integration with BART improves the performance from **representation perspective**. As we know, sentence permutation is one of the pre-training tasks of BART, which makes the sentence representations learned with BART is more compatible with sentence ordering than BERT and demonstrates superior performance. It also could be replaced by other more powerful representation learning model in the future. While exclusive loss contributes to the model from the **aspect of optimization constraint**, aiming to match the positions and sentences one-by-one and alleviate repetition matching. These two techniques **can be combined to work together** to boost the performance, which is what NAON-BART (in Table 1) does. Thanks again for the valuable comments.
>
> **Question_3:** The ChatGPT experiments are interesting but limited. Can you provide more details on the prompt engineering process and lessons learned on how to prompt sentence order effectively?
>
> **Answer_3:** Thanks for the review. As shown in Figure 5-8 (page13-16) and Table 4, we have explored the input format of unordered sentences, e.g., **List** and **Dict**, and observed sentences with Dict could significantly decrease the error rate of wrongly counting of sentences and achieve better performance. For effectively prompting, we have revised and improved the prompts via several ways, which also described in Section Appendix A. For the lessons and tricks learned online, we mainly learn how to effectively prompt the ChatGPT in general way. Such as we learned that the main goal of prompt engineering is to make the task description more precise, concise, and easy to understand for ChatGPT, and make the input and output clear and definite. We then applied the prompt engineering and tricks for sentence ordering by considering the task characteristic. Limited by the prompt engineering, the performance of ChatGPT has not met the expectation, and we will keep exploring on this.

---

### Official Review · Reviewer_7JW7 · 2023-08-11

**Soundness:** 3

**Excitement:**

4: Strong: This paper deepens the understanding of some phenomenon or lowers the barriers to an existing research direction.

**Paper Topic And Main Contributions:**

The paper introduces a new technique for sentence ordering tasks. Specifically, the primary novelty of the work involves the usage of non-autoregressive decoders. As the output length is known exactly beforehand, it makes the use of non-autoregressive decoders ideal for this task. The paper encodes sentences independently, passes the sentence representations to another self-attention based transformer and then uses non-autoregressive decoding. The paper also introduces a greedy inference technique to reduce repetitions in the output. The proposed approach achieves great results along with new SOTA for the task over 6 different datasets.

**Questions For The Authors:**

Please respond to the weakness mentioned above.

**Reasons To Accept:**

1. The paper is mostly well-written.
2. The paper achieved great results beating the previous SOTA Re-BART, which is hard to beat and held the SOTA for nearly 2 years.
3. The utility of non-autoregressive decoders is novel and well-suited as the output length is known in advance.

**Reasons To Reject:**

1. The described approach in the methods section does not achieve the best results. Only when non-autoregressive decoders are applied to BART it achieves the SOTA. It begs the question is the application of the Basic and Contextual Sentence Encoder really needed? Apart from the pre-training task, I feel it helps the model to encode all the sentences together. It would be good to see some empirical evaluation when you train an encoder that takes entire sentences as input either from scratch or using another pre-trained model apart from BART.
2. The paper needs to provide some more introduction to non-autoregressive decoding. Without that it can be a bit difficult for the reader to go through Section 2.4 and 2.5.
3. The current state-of-the-art Re-BART was not mentioned at all in the introduction and related works while describing generation based models. Some of the statements  made in the paper (Line 241-243) when you consider systems like Re-BART.
4. Table 2 in the paper is not complete. The paper hides several baselines that outperform it. These are reported in the Re-BART paper, and  the paper should report them for completeness even though their results are slightly weak.

**Reproducibility:**

3: Could reproduce the results with some difficulty. The settings of parameters are underspecified or subjectively determined; the training/evaluation data are not widely available.

**Reviewer Confidence:**

4: Quite sure. I tried to check the important points carefully. It's unlikely, though conceivable, that I missed something that should affect my ratings.

---

> ### Author Rebuttal · Authors · 2023-08-28
>
> We thank and appreciate the reviewer's careful and valuable comments. To make our paper clearer, below we try to answer the questions and issues raised by the reviewer. Hope it could address the concerns of the reviewer. Thanks very much.
>
> **Question_1:** The described approach in the methods section does not achieve the best results. Only when non-autoregressive decoders are applied to BART it achieves the SOTA. It begs the question is the application of the Basic and Contextual Sentence Encoder really needed? Apart from the pre-training task, I feel it helps the model to encode all the sentences together. It would be good to see some empirical evaluation when you train an encoder that takes entire sentences as input either from scratch or using another pre-trained model apart from BART.
>
> **Answer_1:** Thanks for the constructive review. We think the improvement of integration with BART (NAON-BART in Table 1) mainly comes from the powerful and compatible representation learned from BART pre-trained model. Because **sentence permutation** is one of the pre-training tasks of BART, the representation extracted from BART is quite suitable for sentence ordering and gains significant improvement. To verify the real contributions of Basic and Contextual Sentence Encoder, we have conducted experiments with LSTM-based models, **contextual LSTM** (the same as LSTM-Ptr in Table 1) and **all sentence together LSTM** (we use TogLSTM to denote it), on SIND. The results are as below:
>
>
> $\begin{matrix}
> \hline
> \text{Method} & \text{Acc} & \text{PMR} & \text{$\tau$} \\\\ \hline
> \text{LSTM-Ptr} & 45.33 & 12.7 & 0.47 \\\\
> \text{TogLSTM} & 42.55 & 10.35 & 0.44 \\\\ \hline
> \end{matrix}$
>
> From the results, we observe the LSMT-Ptr with Basic and Contextual Sentence Encoder outperforms TogLSTM with large margin, which verifies the effectiveness of our Basic and Contextual Sentence Encoder. Due to the time limit, we only test with simple LSTM on SIND, we will conduct extensive experiments with pretrained models and other methods on other datasets to comprehensively evaluate the effectiveness of the Basic and Contextual Sentence Encoder. As our NAON is the first attempt at using non-autoregressive decoding for sentence ordering, we have not tried many advanced techniques at current stage. In future work, we will explore more advanced techniques and settings, e.g., the reviewer suggested taking all the sentence as input (but we think the sentences should be order-independent) and exploring relative order information, to further improve the non-autoregressive decoding for sentence ordering. We also hope our work could bring some insights to the community and encourage more attempts at NAT for sentence ordering. Thanks again for the suggestive comments.
>
> **Question_2:** The paper needs to provide some more introduction to non-autoregressive decoding. Without that it can be a bit difficult for the reader to go through Section 2.4 and 2.5.
>
> **Answer_2:** Thanks for the suggestion. To provide more introduction to NAT and make Sec2.4 and 2.5 more friendly to the readers, we will revise corresponding parts with two folds: 1) add a brief introduction to NAT in Line 256 as:
> ```“Our NAD is designed based on a Non-Autoregressive Transformer (NAT) decoder in machine translation, which removes the autoregressive connections between steps and generates all the target words in parallel, rather than step-by-step (i.e., word-by-word).”```
>  and 2) add a subsection of literature review of Non-Autoregressive Transformer (NAT) in Related Works in the revision. We thank again for your helpful suggestion.
>
> **Question_3:** The current state-of-the-art Re-BART was not mentioned at all in the introduction and related works while describing generation-based models. Some of the statements made in the paper (Line 241-243) when you consider systems like Re-BART.
>
> **Answer_3:** We apologize for missing description about RE-BART. We will introduce it in suitable place in the introduction and related works in revision. For the statement in L241-243, we apologize for the imprecise description. Actually, we want to express “recurrent-based”, of which RNN-based model is the most straightforward one. Thanks for the reviewer's reminder, we note that many works employ Transformer for generation, which also applies autoregressive (recurrent) decoding strategy and is excluded by the description of ‘RNN-based’. We will revise this in the revision to make the description more precise.
>
> **Question_4:** Table 2 in the paper is not complete. The paper hides several baselines that outperform it. These are reported in the Re-BART paper, and the paper should report them for completeness even though their results are slightly weak.
>
> **Answer_4:** Thanks for the constructive suggestions. We are not intentionally to omit the baselines. We actually have conducted the experiments of BERSON, NAON-RO, RE-BART, and NAON-BART, but we omitted them due to the page limitation. As mentioned in Answer_1, our NAON is a prior attempt for sentence ordering with non-autoregressive decoding, and we only use the vanilla NAD without advanced techniques to fairly compare with baselines. Then we have also verified its generalization ability by integrating with BERSON and BART. Therefore, we chose to remove these results from Table 2 to meet the page requirements. To dispel your doubts, we present the results below and will include them in Table 2 in the revision.
>
> $\begin{matrix}
> \text{Model} & \text{arXiv (head)}  & \text{arXiv (tail)} & \text{SIND (head)} & \text{SIND (tail)}           \\\\ \hline
>     \text{BERSON}  & 94.75  & 76.69  & 84.95  & 64.87  \\\\
>  \text{NAON-RO} & 94.83  & 76.71  & 84.98  &  64.91 \\\\ \hline
>  \text{RE-BART} & 96.46 & 80.62 & 87.97 & 73.02 \\\\
>  \text{NAON-BART} & 98.98 & 85.91 & 90.1 & 80.99 \\\\
>      \hline
> \end{matrix}$
>
> From the results, we observe that equipped with our non-autoregressive decoder, NAON-RO and NAON-BART ourperform the corresponding autoregressive ones in head and tail prediction, which are consistent with the results and conclusions in Table 1.

---

### Meta-Review · Area_Chair_y9fd · 2023-09-19

**Recommendation:** 3

**Metareview:**

This paper is about a new method for sentence ordering. Although the motivation and the task are not the most "exciting" (sentence ordering as a proxy for discourse coherence), the reviewers agree on the novelty of the method which predicts sentences in parallel (as opposed to sequentially). One reviewer expressed some concern that the task is too simple which risks not pushing the method enough, but they all agree that the results are "sound" and with potential to be built upon by other researchers working on non-autoregressive generation.

---

### Decision · Program_Chairs · 2023-10-07

**Decision:**

Accept-Findings

**Comment:**

This paper is about a new method for sentence ordering. Although the motivation and the task are not the most "exciting" (sentence ordering as a proxy for discourse coherence), the reviewers agree on the novelty of the method which predicts sentences in parallel (as opposed to sequentially). One reviewer expressed some concern that the task is too simple which risks not pushing the method enough, but they all agree that the results are "sound" and with potential to be built upon by other researchers working on non-autoregressive generation.